# Hepatitis C Viral Replication Complex

**DOI:** 10.3390/v13030520

**Published:** 2021-03-22

**Authors:** Hui-Chun Li, Chee-Hing Yang, Shih-Yen Lo

**Affiliations:** 1Department of Biochemistry, Tzu Chi University, Hualien 97004, Taiwan; huichun@gms.tcu.edu.tw; 2Department of Laboratory Medicine and Biotechnology, Tzu Chi University, Hualien 97004, Taiwan; cheehing2@gms.tcu.edu.tw; 3Department of Laboratory Medicine, Buddhist Tzu Chi General Hospital, Hualien 97004, Taiwan

**Keywords:** hepatitis C virus, replication organelles, NS3 to NS5B proteins, direct-acting antivirals

## Abstract

The life cycle of the hepatitis C virus (HCV) can be divided into several stages, including viral entry, protein translation, RNA replication, viral assembly, and release. HCV genomic RNA replication occurs in the replication organelles (RO) and is tightly linked to ER membrane alterations containing replication complexes (proteins NS3 to NS5B). The amplification of HCV genomic RNA could be regulated by the RO biogenesis, the viral RNA structure (i.e., cis-acting replication elements), and both viral and cellular proteins. Studies on HCV replication have led to the development of direct-acting antivirals (DAAs) targeting the replication complex. This review article summarizes the viral and cellular factors involved in regulating HCV genomic RNA replication and the DAAs that inhibit HCV replication.

## 1. Introduction

Infection with the hepatitis C virus (HCV) can cause chronic hepatitis C (CHC), liver cirrhosis, hepatocellular carcinoma, and other extra-hepatic manifestations. The prevalence of CHC patients worldwide was around 71 million in 2017 (https://www.who.int/hepatitis/publications/global-hepatitis-report2017/en/). HCV belongs to the family Flaviviridae and genus *Hepacivirus*. Its genome is a single-stranded RNA with positive polarity. Many different but closely related circulating HCV variants (i.e., quasispecies) can be detected in CHC patients due to the low fidelity of the HCV RNA polymerase (NS5B) and its high replication rate [1]. Thus, HCV genomic RNA sequences are highly heterogeneous among different isolates. At present, HCV is classified into at least six major genotypes (GT 1 to 6) [2,3]. The geographic distribution of different HCV genotypes varies [3]. Subtype 1a is found throughout the US and Northern Europe, while subtype 1b is widely distributed throughout the world and is a major subtype in Japan. Genotype 2 is present in the same areas as genotype 1. Subtype 3a is widely distributed in South Asia and Oceania, while subtype 3b is mainly found in East Asia. Genotype 4 is mainly present in the Middle East, Northern to Central Africa, and Europe. Subtype 5a is mainly found in South Africa. Genotype 6 is mainly distributed throughout East and South-East Asia.

The life cycle of HCV begins with its binding to cells. Numerous cellular factors, including proteins, lipids, and glycans, promote the entry of HCV particles into hepatocytes. HCV initially attaches to the surface proteoglycans, e.g., the scavenger receptor BI, and to the tetraspanin CD81. After lateral translocation to tight junctions, claudin-1 and occludin proteins become essential for HCV entry. HCV particles are engulfed by clathrin-mediated endocytosis and then fused with endosomal membranes in low-pH conditions. Viral genomic RNA is then released into the cytoplasm [4]. Then, the HCV genomic RNA is used for both protein translation and viral RNA replication. HCV RNA replication takes place within the replication organelles (RO) in the endoplasmic reticulum (ER). Finally, HCV utilizes the biosynthetic pathway of very-low-density lipoprotein to assemble the viral particles and egress from the cells [5].

The HCV RNA genome (~9600 nucleotides) possesses one open reading frame that is flanked by 5’ and 3’ untranslated regions (UTRs) (Figure 1a). Translation of the viral RNA leads to the synthesis of a polyprotein, which is processed into individual viral proteins via cleavages of both cellular and viral proteases. The structural proteins (i.e., the core and envelope glycoproteins E1 and E2) are the main constituents of HCV particles, whereas the viroporin p7 and nonstructural protein 2 (NS2) are involved in virion assembly [6]. The remaining nonstructural proteins (i.e., NS3, NS4A, NS4B, NS5A, and NS5B; NS3-NS5B) that have specific roles in viral genome amplification form the replication complex [7,8,9]. The roles of different viral proteins in HCV replication are summarized in Table 1.

Over thirty years of research on the mechanisms of HCV replication has led to the successful development of direct-acting antivirals (DAAs) targeting the replication complex [10]. We summarize the viral and cellular factors involved in regulating HCV genomic RNA replication and the DAAs inhibiting HCV replication in this review article.

## 2. Viral Replication Organelles (RO)

HCV induces cellular membrane alterations referred to as the membrane web (MW) for viral RNA replication [11]. Different types of membrane alterations induced by HCV were observed [12,13]. Among these membrane alterations, double-membrane vesicles (DMVs) induced by HCV infection associated with double-stranded RNA (dsRNA) and nonstructural proteins are believed to be the sites of viral genome replication (i.e., viral replication organelles (RO)) in cultured cells (Figure 2). DMVs comprise the predominant HCV-induced membrane structure that forms in the cytoplasm close to the lipid droplets (LDs) in cultured cells. LDs with HCV core and NS5A proteins surrounded by ER is close to the HCV replication (e.g., DMV) and assembly sites. HCV genomic RNA synthesized in the DMVs is transferred by HCV nonstructural proteins and encapsidated by the core proteins to form the nucleocapsid. The HCV nucleocapsid will then interact with glycoproteins E1/E2 in the assembly sites and bud into the ER lumen [14]. DMVs are heterogeneous in size, with an average diameter of ~200 nm. At late time points after infection, multi-membrane vesicles were observed and believed to reflect a stress response induced by high-level virus replication [13,15,16]. These HCV-induced DMVs are morphologically similar to those identified in cells infected with coronaviruses, picornaviruses and noroviruses [17]. Previous studies also showed that HCV could induce membrane alterations in the hepatocytes of HCV-infected patients [18,19].

HCV-induced single-membrane vesicles (SMVs) were also detected sporadically in cultured cells [13,15,16]. Unlike observations from the cultured cells, a recent report showed that the MW detected in liver tissues of HCV-infected patients seems essentially to be made of clusters of SMVs [20]. Further studies are needed to clarify this issue.

The majority of HCV DMVs appear to be closed structures, and only a few of them have an opening pore toward the cytosol [13]. It is not yet known whether HCV RNA replication takes place on the interior or exterior membrane surface of the DMVs. If HCV RNA replication occurs on the interior surface of DMVs, then a transport mechanism must be present to allow the influx of metabolites (e.g., nucleoside triphosphates) required for replication and the exit of newly synthesized viral RNAs for translation or virion assembly [8]. This hypothesis is supported by the findings that HCV hijacks specific cellular components responsible for nucleocytoplasmic transport and that these cellular factors are probably involved in maintaining a transport system between the cytosol and the interior of viral ROs [21,22].

There are several advantages to forming viral ROs for HCV RNA synthesis [17]. First, the viral replication complex (NS3-NS5B) and cellular factors responsible for HCV RNA replication can be concentrated in ROs. Second, ROs, by excluding cellular RNAs, contribute to the template specificity of the replication complex. Third, the replication intermediates (i.e., dsRNA) can be protected from the detection of cellular innate immune sensors. Fourth, ROs facilitate the separation of different stages in the life cycle of HCV (translation vs. replication, replication vs. assembly) by compartmentalization [22]. Fifth, several reports showed that viral RNA and proteins associated with the viral ROs are protected from cellular proteases and nucleases, indicating that RNA replication occurs in a membranous environment separated from the surrounding cytoplasm [16,23,24].

**Figure 2 viruses-13-00520-f002:**
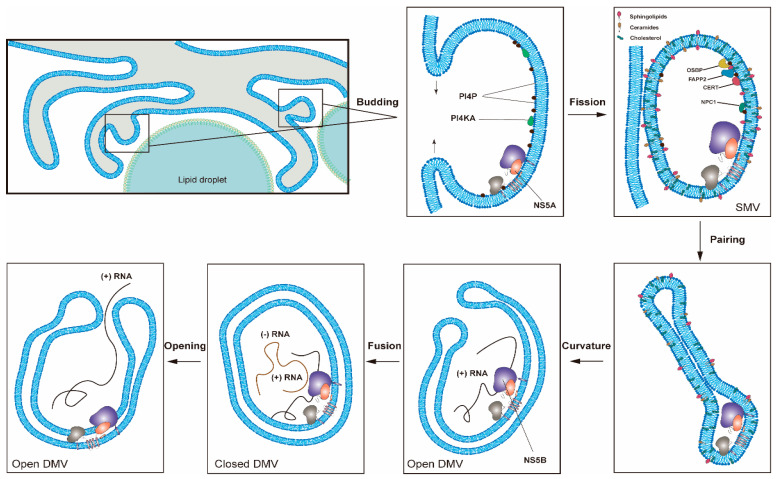
One proposed model for the formation of HCV replication organelles (RO). Double-membrane vesicles (DMV) biogenesis is a complex process possibly requiring several membrane remodeling steps, including budding, fission, pairing, curvature, and fusion [11,17]. HCV DMV formation can be induced by NS5A with the help of other nonstructural proteins [25,26]. HCV activates the lipid kinase PI4KIIIα to generate enhanced levels of PI4P, which in turn attracts lipid transport proteins (e.g., OSBP, FAPP2, NPC1 and CERT) delivering cholesterol and glycosphingolipids into DMVs. PSTPIP2, a protein with membrane-deforming activity, and PLA2G4C are also critical for membrane web (MW) formation. HCV RNA replication can be conducted by NS5B with the help of other nonstructural proteins in the closed DMVs. After the completion of HCV genomic RNA synthesis, the newly synthesized viral RNAs will be released from the open DMVs (possibly with Kaps or Nups) for translation or virion assembly. Other possible models for DMV formation have also been proposed [11].

### 2.1. HCV Proteins Involved in RO Formation

Although any protein of the HCV replication complex (NS3-NS5B) can induce membrane alterations, NS5A is the only one capable of inducing DMV formation [13,25,26]. The insertion of the amino-terminal amphipathic α-helix of NS5A into just one membrane layer could facilitate the membrane curvature required for RO formations [27]. The observation that NS5A inhibitors (e.g., daclatasvir) block the HCV RO formation independent of RNA replication demonstrated the essential role of NS5A in the RO formation [28]. The efficiency of the DMV formation induced by NS5A alone is low but is greatly enhanced when the other nonstructural proteins are also expressed, e.g., NS4B, NS3, or NS5B [26,29,30].

Similar to NS5A, NS4B also contains terminal amphipathic α-helices, which can alter membrane properties. Moreover, the membrane topology of NS4B is likely to undergo posttranslational changes [31,32,33,34], possibly in an NS5A-regulated manner [35]. Moreover, NS4B contains a GXXXXGK P-loop for nucleotide triphosphate binding, which may be involved in membrane rearrangements [36]. In addition, NS4B could form homo-oligomeric complexes, which are required for the RO formation [29,37,38].

### 2.2. Cellular Factors Involved in HCV RO Formation

In addition to having direct involvement in the RO formation, viral proteins also contribute to membrane alterations by recruiting cellular factors required for RO biogenesis. For example, NS5A. Cyclophilin A (CypA), receptor for activated protein C kinase 1 (RACK1) and ATG14L were found to participate in DMV formation for HCV replication by interacting with NS5A [39,40,41,42], while Surf4 and prolactin regulatory element-binding (PREB) participated by interacting with NS4B [43,44]. PSTPIP2 (Proline-serine-threonine phosphatase interacting protein 2) with membrane-deforming activity and PLA2G4C (cytosolic phospholipase A2 gamma) are also important for HCV RO formation via direct interactions with NS4B and NS5A [45,46].

The HCV replication complex is reportedly associated with membrane lipid micro-domains (i.e., lipid rafts) [47,48], enriched with cholesterol, sphingolipids, and certain proteins. Lipid rafts generally contain three to five times the cholesterol content found in the surrounding bilayer [49]. Shaping an ER membrane into an RO for HCV RNA replication requires not only viral and cellular proteins but also lipid synthesis [50,51,52]. Multiple reports have indicated that HCV modulates lipid metabolism (e.g., cholesterol and fatty acid biosynthesis) to promote viral replication [53,54,55]. This modulation results in de novo lipid biosynthesis in order to increase the membrane surface area required for the RO formation. SREBPs (the sterol regulatory element-binding protein) are major regulators of lipid metabolism and major transcription factors for the expression of genes required for lipid biosynthesis [56]. HCV NS4B has been shown to activate SREBP, leading to the elevated transcription of genes involved in lipogenesis, e.g., fatty acid synthase (FASN) [57].

Modulation of the lipid environment of RO via HCV also includes the recruitment and activation of the lipid kinase PI4KIIIα by NS5A and NS5B proteins to generate enhanced levels of phosphatidylinositol 4-phosphate (PI4P) at the RO [58]. PI4P could attract lipid transport proteins (oxysterol-binding protein (OSBP), four-phosphate adaptor protein 2 (FAPP2), NPC1, and ceramide transfer protein (CERT) to deliver glycosphingolipids, cholesterol, and ceramide to RO, respectively [16,59,60]. Recently, it was reported that HCV NS3/4A protease controls the activity of 24-dehydrocholesterol reductase (DHCR24), catalyzing the conversion of desmosterol to cholesterol and regulating the lipid environment for HCV RNA replication [61]. In contrast, cholesterol-25-hydroxylase induced by interferon could block MW formation via the production of 25-hydroxycholesterol and thus restrict HCV replication [62]. Recently, C19orf66, an interferon-stimulated gene, was reported to inhibit HCV by preventing the elevation of PI4P and altering RO formation [63].

In addition to these cellular factors, several studies have shown that autophagy plays an early role in establishing HCV replication [64,65,66]. DMVs induced by HCV accumulated at the MW are morphologically similar to autophagosomes [15]. Thus, autophagy may help to induce MW formation during HCV replication [67]. However, DMVs induced by HCV with an average diameter of ∼200 nm are smaller than autophagosomes of 500 to 1000 nm in diameter. The exact role of autophagy in HCV RO formation requires further investigation [68].

Proteins in the nuclear transport machinery (including soluble nuclear transport factors (NTFs), e.g., karyopherins (Kaps)) and nucleoporins (Nups) in the nuclear pore complexes (NPCs) are probably involved in the transfer between the cytosol and the viral ROs [21,22].

## 3. Genome Replication

HCV genome replication requires at least viral genomic RNA and viral NS5B protein (RNA-dependent RNA polymerase; RdRp) in the ROs. Indeed, NS5B can de novo initiate and copy HCV genomic RNA without the help of other factors in vitro [69,70,71]. HCV genome replication could be modulated by the HCV RO biogenesis, viral RNA structure (i.e., cis-acting replication elements), viral proteins (particularly, NS5B), and other cellular factors [7].

### 3.1. HCV RNA Elements Involved in the Genome Amplification

Functional RNA structures have been identified throughout the HCV genome [72]. The secondary RNA structural elements found in both the positive-strand viral genome and the negative-strand replication intermediate are important for viral genome amplification. The RNA structures found in the 5’UTR of the positive-strand viral genome are primarily involved in translation initiation, but several stem-loop (SL) structures have been associated with genome replication [73]. This association is attributed to the SL elements in the 3’ end of the negative-strand RNA, which forms secondary structures distinct from those found in the positive strand [74,75].

In addition to the 5’ end structural elements of positive-strand RNA, several RNA elements found in the core coding region, the 3’UTR, and the NS5B-coding region are also essential for genome replication (Figure 1a) [76,77,78,79]. The 3’UTR is comprised of a variable region, a poly(U/UC) tract, and a highly conserved 3’X-tail [80,81]. The X-tail contains three SL structures [82]. All of these three SLs are essential for RNA replication [83] and barely tolerate any mutations [84,85,86]. X-tail is probably the main regulatory element for the initiation of negative-strand synthesis. Initiation of negative-strand RNA synthesis starts at the terminal uridine that is base-paired to guanosine in the 3’X SL1 [87]. Several long-range RNA–RNA interactions between the 3’ and 5’ elements, facilitated by trans-acting cellular factors, have been shown to potentiate viral genome replication [88,89,90].

### 3.2. HCV Proteins Responsible for Genome Replication

HCV RNA replication depends on the specific cis- and trans-acting activities of HCV nonstructural proteins (NS3-NS5B) [91,92]. Recently, NS3-NS5B proteins in RO have been visualized and analyzed using super-resolution microscopy [93].

NS5B (HCV RNA-dependent RNA polymerase) encompasses an amino-terminal catalytic domain that makes up the majority of NS5B, followed by a linker sequence and a C-terminal transmembrane domain (TMD) tethering the catalytic domain to the membrane (Figure 3) [69,94,95]. The TMD is essential for RNA replication in cells yet dispensable for enzymatic activity in vitro. Like all other viral RdRps, NS5B has a “right-hand” shape containing palm, thumb, and fingers subdomains [96]. In addition, HCV NS5B contains a β-flap domain-specific to the RdRps of the Flaviviridae family and a linker domain common to de novo initiating enzymes. Each of these domains contributes to the specific steps in viral RNA synthesis [96]. Regulation at the N-terminal finger subdomain of NS5B through phosphorylation has been demonstrated [97].

The HCV RNA synthetic process conducted by NS5B can be divided into four steps: RNA binding, initiation, elongation, and termination [7]. Structural evidence indicates that NS5B uses de novo initiation to replicate the HCV RNA genome in cells [69,94,98]. Moreover, NS5B is capable of internal initiation via functional replication and likely only requires terminal initiation in its natural environment [99]. It is believed that initiation begins at the 3’ end of the HCV genomic RNA and requires high levels of GTP that bind to an allosteric site in the NS5B (β-flap domain) to act as structural support to prime the initiation step [100,101]. The 3’ end of the positive-strand RNA is a poor template for de novo initiation, as it is concealed within an SL of the X-tail. In contrast, the 3’ end of the negative-strand RNA consists of an SL with an overhang that serves as a highly efficient initiator of RNA synthesis. This difference likely contributes to the 10-fold excess of positive- over negative-strand RNA. HCV NS5B protein alone does not seem to have specificity for the viral genome. However, studies showed that interactions between NS3 helicase, NS5A, and NS5B are required for initiation of RNA synthesis, which indicates that template specificity is conferred by a combination of these viral factors [102].

The closed conformation observed in the crystal structures of NS5B most likely represents the initial state of the enzyme [101,103]. The switch from initiation to elongation seems to be one of the rate-limiting steps, and GTP facilitates this switch [104]. Excess primers are synthesized before NS5B continues to elongate RNA synthesis [104]. A major conformational change towards an open conformation of NS5B is required in this step, probably driven by the removal of the linker sequence to allow the exit of the dsRNA [105]. Residue 405 in the thumb of NS5B seems to be critical for efficient primer synthesis, for switching from initiation to elongation, and for replication efficiency [106,107]. After the initiation of HCV RNA replication, NS5B elongates nascent RNA synthesis by 100 to 400 nucleotides per minute and can copy an entire RNA genome in vitro [70,71,103,108]. NS5B protein–protein interactions are also important for the initiation and elongation of RNA de novo synthesis [109]. It is not yet known how HCV RNA synthesis is terminated.

HCV NS5B is believed to be remarkably error-prone and leads to an error rate of ~10^−4^ per site in every round of replication, with a strong bias towards G:U/U:G mismatches [110,111]. However, NS5B has also been reported to have a nucleotide excision mechanism, which may allow limited error correction [112]. Indeed, a recent report showed that the calculated fidelity of NS5B ranges between 10^−4^ and 10^−9^ for different mismatches [113].

In addition to NS5B, other viral and cellular proteins also contribute substantially to HCV RNA synthesis. NS5B recruits NS3 to facilitate processive elongation of RNA synthesis [114]. NS3 contains a carboxy-terminal DExD-box helicase domain (NS3h) and an amino-terminal protease domain that functions in conjunction with the cofactor, NS4A (Figure 4). The NS3 protease domain is responsible for HCV polyprotein processing and also contributes to the activity of the helicase domain through an allosteric mechanism [115,116,117]. Specific mutations in the NS3 helicase domain modulating the nucleic acid unwinding activity, which also affect HCV’s replicative ability, indicate a role of NS3h in HCV RNA synthesis [118,119,120]. The NS3h may play roles in (i) resolving strong stem-loop structures at the 3’ end of the genome to facilitate initiation of RNA synthesis by NS5B; (ii) unwinding replication intermediates (i.e., dsRNA) during RNA synthesis to support NS5B in the elongation phase; and (iii) striping proteins off the RNA or delivering RNA for packaging into virions via the process of ssRNA translocation [105,121].

In addition, the linker sequence between the protease and helicase domains of NS3 has been suggested to have a regulatory role in replication and assembly [122].

NS4A is only 54 amino acids (a.a.) long and contains three domains, an N-terminal transmembrane domain, a central NS3-interacting domain, and a C-terminal domain (Figure 4) [123,124]. The N-terminal transmembrane α-helix of NS4A anchors NS3 to intracellular membranes. The central domain acts as the NS3 protease cofactor and also interacts with cellular creatine kinase B, which has been reported to augment NS3 helicase activity and HCV replication [125]. The C-terminal domain contains a kink region and an acidic region, which is required for viral assembly and envelopment [126]. The interactions between NS4A and NS4B also control HCV RNA replication [126].

NS4B contains two amino-terminal amphipathic α-helices, followed by four transmembrane spanning α-helices, and another two carboxy-terminal amphipathic α-helices [127]. In addition to supporting RO formation, NS4B also plays a role in virus replication. The S/T cluster and GXXXG motif in the first and second transmembrane segments of NS4B are important for virus replication [128]. NS4B also forms oligomeric complexes via self-interaction, which is required for HCV RNA replication, in addition to the above-mentioned RO formation [30,37,38]. The interaction between the NS4B and NS5A is reportedly involved in viral replication [129,130].

The nonenzymatic NS5A protein is a multifunctional zinc-binding phosphoprotein involved in different stages of the HCV life cycle, including replication, assembly, and egress [131]. NS5A has a length of ~450 amino acids and contains an N-terminal amphipathic α-helix, which is critical for its membrane targeting [132], and three domains (domains I, II and III) separated by two low-complexity sequences (LCS) (Figure 5a). Domain I (a.a. 33–213) is an RNA-binding region linked to virus replication as well as the aforementioned RO biogenesis and probable functions in several alternative dimerized states [26,133,134]. In addition to its role in HCV RNA replication, domain I of NS5A may have a role in virus assembly [135]. Domains II (a.a. 250–342) and III are predicted to be largely unstructured and interact with viral and/or cellular factors, including cyclophilin A and phosphatidylinositol 4-kinase IIIα (P I4KA) [136,137]. Domain III (a.a. 356–447) functions primarily in virion assembly [138]. NS5A has been reported to help NS5B bind to the HCV RNA template [114].

Different NS5A functions seem to be regulated through differential phosphorylation states [139,140,141,142,143,144]. Many studies have linked several phosphorylation sites in the LCS region to viral genome replication and show that reducing the phosphorylation of these sites by blocking the casein kinase I isoform α (CKIα) suppresses HCV replication and virion assembly [140,145,146,147,148,149]. These results suggest that phosphorylation in the LCS region may function as a regulatory switch between RNA replication and virion assembly. The exact mechanisms for how this multifunctionality is achieved are largely unknown and may be genotype-dependent [150]; however, it is thought that various NS5A phosphor-variants bind to distinct cellular factors, e.g., CypA, P I4KA, VAP A/B, or apolipoprotein E, and then exert different functions [7].

### 3.3. Cellular Factors Involved in HCV Genome Replication

Cellular factors, including lipids, miRNAs, and proteins, are involved in HCV replication [11,151,152]. Lipidomic analysis has revealed distinct alterations of cellular lipid composition via HCV infection [55,153]. The inhibition of fatty acid synthesis by blocking acetyl-CoA carboxylase decreases viral replication [54]. Furthermore, an increase in saturated and monounsaturated fatty acids augmented HCV replication, while an increase in polyunsaturated fatty acids suppressed replication [54]. Distinct lipids may form lipid rafts required for assembly and viral replication activity. Indeed, the removal of cholesterol from the replication complex impairs its activity [16]. In addition, sphingolipids (e.g., sphingomyelin) were shown to stimulate HCV replication activity [154,155]. A recent report further demonstrated that sphingomyelin is essential for the structure and function of HCV DMVs [156]. Lipid metabolism is regulated by a family of sterol regulatory element-binding proteins (SREBPs), which are transcription factors controlling the expression of more than 30 lipogenic genes. HCV-induced ER stress or viral proteins (e.g., NS4B) could trigger the activation of SREBPs [56,57]. Moreover, the 3’UTR of HCV genomic RNA could bind to cellular RNA helicase DDX3X, which acts as an intracellular sensor to induce SREBP expression [157]. Although the effect of HCV infection on cellular lipid metabolism is known, further studies are still needed to clarify how distinct lipids contribute to HCV RNA replication.

In addition to lipids, liver-specific miR-122 could bind to the two adjacent sites of HCV 5’ UTR (Figure 1a) [158], forming a ternary complex [159]. Through this interaction, in addition to stimulating translation [160], miR122 could also facilitate RNA replication [161] by protecting the genome from cellular DUSP11 pyrophosphatase activity [162] and subsequent degradation by the exonucleases Xrn1 [163,164] and Xrn2 [164,165].

Many cellular proteins are involved in regulating HCV replication [7,8,105], and only those playing a direct role in HCV RNA replication are mentioned here. Several cellular proteins could facilitate genome circularization and enhance RNA replication by binding to the 5’- and the 3′-UTR of viral RNA. These proteins include La [166], hnRNP L [167], the NFAR protein complex (NF90, NF45, and RHA) [168], PTB [169], PCBP2 [168] and RNA binding protein 24 [170]. High-mobility group box 1 (HMGB1) interacting with SL 4 of 5’-UTR [171], Src-associated in mitosis 68-kDa (Sam68) protein binding with SL 2 of 5’-UTR [172], and heat shock cognate protein 70 (Hsc 70) interacting with poly-U/UC in the 3’-UTR [173] could also promote HCV replication.

Several cellular factors enhance HCV RNA replication via interaction with NS5B, including cellular chaperonin TRiC/CCT [174], ribonucleotide reductase M2 (RRM2) [175], HuR [176], VAPB-MSP [177], CYP4F12 [178], and fatty acid synthase [179].

DDX3 [180], Y-box binding protein 1 (YB-1) [180], FKBP6 [181], and human choline kinase-α (hCKα) [182,183] could interact with NS5A to facilitate HCV RNA replication. Cellular Cyp A [137,184] and human replication protein A (RPA) [114] could bind to NS5A and stimulate the binding of NS5A to NS5B and viral RNA to facilitate HCV RNA replication.

The viral NS3 protein is also an important component of the HCV replication complex. Rad51 [185] and GBF1 [186] could interact with NS3 and promote HCV RNA replication. Rab (the Ras superfamily of small GTPases) 5 and 7 colocalize with NS4B, and Rab2, 5, and 7 are required for HCV RNA replication [187]. Both VAP-A and VAP-B, enriched in purified DMVs [16], and valosin-containing protein (VCP) [188], interact with NS5A and NS5B and assist in the formation of the replication complex [189,190].

## 4. Direct-Acting Antivirals (DAAs)

A combination of pegylated interferon (IFN) alpha and ribavirin was used to treat HCV-infected patients before 2011 [2,10]. IFN inhibits viral replication by inducing more than 300 interferon-stimulated genes (ISGs) [191]. Indeed, IFN-alpha was reported to inhibit multiple steps of the HCV life cycle, leading to a reduction in viral protein synthesis and eventually suppression of viral RNA amplification [192]. The precise mechanism(s) by which ribavirin exerts its anti-HCV activity is not fully understood. The antiviral activity of ribavirin probably occurs via a combination of different mechanisms [193]: (1) immunomodulation, (2) modulation of ISG expression, (3) inhibition of inosine 5’-monophosphate dehydrogenase by ribavirin 5’-monophosphate, (4) inhibition of eIF4E, (5) inhibition of the HCV RdRp by ribavirin 5’-triphosphate, and (6) induction of viral mutagenesis.

Understanding the HCV proteins essential for HCV replication has enabled the development of DAAs targeting viral proteins (NS3 protease activity, NS5A protein and NS5B polymerase) [10,194].

Structural investigations on the NS3/4A protease have facilitated drug developments. The NS3-NS4A protease catalyzes HCV polyprotein cleavage (Figure 1b) [92]. Current NS3/4A inhibitors target the protease activity of NS3/4A. The inhibiting compounds include (I) substrate-derived peptide inhibitors: boceprevir and telaprevir (reversible covalent binders), ciluprevir (P1–P3 macrocycles), BMS605339 (acyl sulfonamide P1) and MK-4519 (P2–P4 macrocycles); (II) second-generation NS3/4a protease inhibitors (faldaprevir, danoprevir, asunaprevir, vaniprevir, simeprevir); (III) novel structural features/classes (sovaprevir and deldeprevir; DX-320; GS-9256 and PHX1766; MK-2748 and vedroprevir; MK-6325; MK-8831); and (IV) third generation NS3/4a protease inhibitors (paritaprevir, grazoprevir, glecaprevir and voxilaprevir) [195].

The current NS5A inhibitors that have been approved for HCV treatment include daclatasvir (DCV), ledipasvir (LDV), ombitasvir (OMV), elbasvir (ELB), velpatasvir (VEL), and pibrentasvir (PIB) [196]. Despite the successful development of DAAs against NS5A, the exact mechanism of these NS5A inhibitors remains largely unknown. Previous studies have shown that NS5A inhibitors bind the dimerized domain I, thereby potentially blocking NS5A to exert its functions (Figure 5b) [28,197]. Indeed, the direct binding of DCV and VEL to the isolated NS5A domain I (a.a. 33–202) and to full-length NS5A has been reported [198,199].

As a key player in HCV RNA synthesis, NS5B is an important target for anti-HCV drug development. Currently, NS5B inhibitors are divided into two categories: nucleoside/nucleotide inhibitors (NIs) and non-nucleoside/nucleotide inhibitors (NNIs) [200]. NIs (e.g., sofosbuvir, SOF) bind to the catalytic pocket of NS5B, while NNIs (e.g., dasabuvir) exhibit noncompetitive mechanisms of action at sites away from the active site. SOF is converted in hepatocytes to an active nucleoside triphosphate form, which competes with uridine triphosphate. Incorporation of the active nucleotide analogs into newly synthesized RNA by NS5B was resistant to excision [201] and would block the closure of the NS5B active site upon the binding of the next correct incoming NTP, which prevented further nucleotide addition [202]. NNIs bind to one of five allosteric sites located on either the thumb or palm domains of NS5B, interfere with the conformational changes of NS5B and thus block the polymerase activity [203,204,205,206].

## 5. Resistance-Associated Variants (RAV)

Due to the risk of selecting resistance-associated variants (RAV), all-oral interferon-free combined therapies are commonly composed of two or three DAAs inhibiting different targets (NS3 protease, NS5A or NS5B), supplemented with ribavirin if necessary. At present, anti-HCV therapy is considered successful when no virus is detected in the blood 12 weeks after the termination of the treatment (sustained virological response; SVR12) [194,207]. In 2020, approximately 98% of CHC patients were successfully cured using the combination of two or three DAAs for a treatment of 8 to 12 weeks [10].

The effectiveness of DAAs as anti-HCV therapies may be affected by resistance-associated substitutions (RASs) that reduce the viral sensitivity to DAAs and then result in resistance-associated variants (RAVs) [204].

In general, it has been found that RASs present in low proportions (<15%) do not significantly affect treatment outcomes [208]. Thus, in this condition, it is not recommended to identify RASs before the treatment (http://www.hcvguidelines.org). In contrast, testing for RASs prior to the treatment may help to optimize anti-HCV therapy when RASs comprise a proportion of the overall population greater than 15% [208].

Several factors may affect the development of RAVs [204]. First among them is the mutation rate of NS5B polymerase. As mentioned earlier, the calculated fidelity of NS5B is between 10^−4^ and 10^−9^ [113]. The second is the replication rate of the virus. HCV replication is estimated to be 1.3 × 10^12^ virions per day based on the mathematical model [209]. The third is the genetic barrier. RAVs of drugs with a lower genetic barrier require fewer mutations and thus develop more rapidly. The fourth is fitness. RAVs with low levels of fitness do not replicate well without selection (drug treatment) and thus are not detected by commonly used molecular techniques. Finally, there is the pressure of selection. Exposure to suboptimal concentrations of DAAs will result in the selection of RAVs.

Many RASs associated with treatment failure of DAAs have been identified from NS3 regions, and especially from NS5A. The genetic barrier against the NS3/4A inhibitors is relatively low [210]. Luckily, many NS3 RASs to all generations of NS3/4A inhibitors have a low-level fitness [211], except the Q80K mutation (Figure 4a) [212,213]. After the withdrawal of inhibitors, NS3 RASs disappear gradually as the environmental pressure of selection for these RASs is removed. Thus, NS3 RASs are generally found at low proportions (0.1% to 3.1%) in the patients [211,214].

The genetic barrier to NS5A inhibitors is also low [28]. The clinically significant RASs that have been identified include M28A/G/T, Q30D/E/G/H/K/L/R, L31F/M/V, and Y93C/H/N/S, all of which result in high-level resistance to NS5A inhibitors [215,216,217]. M28, Q30, and L31 locate in the linker region between the N-terminal α-helix and domain I, while Y93 is within domain I on the putative dimer interface of NS5A (Figure 5a) [28,218,219]. Due to higher replicative fitness, NS5A RASs persist for years after the removal of inhibitors. Thus, their presence will strongly affect the retreatment outcomes of NS5A inhibitors [204].

The genetic barrier to SOF is very high. Thus, naturally occurring RASs in SOF have rarely been detected in HCV-infected patients. In addition to a high resistance barrier and no detectable pre-existing resistant variants at baseline, SOF also offers pan-genotype coverage. More important, resistance is not an issue for combination therapies with the nucleoside inhibitor. In contrast to SOF, the genetic barrier to NS5B NNIs is relatively low. Thus, the RASs to NS5B NNIs are more commonly detected [220]. As expected, the identity of the RASs that confer resistance to NNIs depends on the particular region targeted by the NNI [204,211].

The advent of DAAs has led to great success in anti-HCV treatment. However, baseline RASs to DAAs may have a significant effect on treatment outcomes in a certain number of HCV-infected patients. Further understanding of the mechanisms of DAA resistance will help to provide better anti-HCV therapy.

## 6. Conclusions

The study of the life cycle of HCV has progressed significantly following the development of in vitro HCV culture systems [221]. Understanding HCV RNA replication has led to the successful development of DAAs targeting NS3, NS5A, and NS5B. HCV will not integrate its genome into cellular chromosomes and thus allowing the curing of HCV infection. To achieve the elimination of HCV infection in 2030, as expected by the World Health Organization, screening for asymptomatic carriers and easy access to the DAAs are very important.

Many questions still remain unanswered regarding HCV RNA replication, such as the sequential events driving viral RO biogenesis, the three-dimensional architecture of the viral replication complex, the termination of RNA replication, and the transfer of the newly synthesized viral RNA. Further understanding of HCV RNA replication will not only provide insight into HCV replication strategies but also will shed light on other positive-strand RNA viruses inducing similar reorganizations of cellular membranes, including coronaviruses, picornaviruses, and noroviruses [17,222]. Further investigation of this topic is still needed.

## Figures and Tables

**Figure 1 viruses-13-00520-f001:**
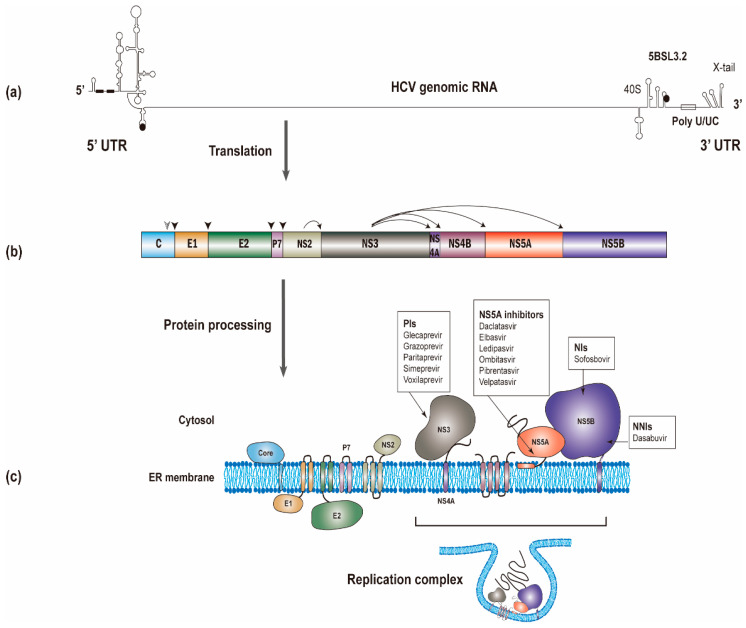
The synthesis of the hepatitis C virus (HCV) proteins. (**a**) The start and stop codons for protein translation were marked by black circles, while two recognition sites on the 5’ UTR for miR-122 were marked by black rectangles. (**b**) The polyprotein is co- and post-translationally cleaved by cellular or viral proteases to yield the structural proteins (core, E1 and E2) and the nonstructural proteins (p7, NS2, NS3, NS4A, NS4B, NS5A and NS5B proteins). The core, E1, and E2 are processed by cellular signal peptidase (filled arrowhead). A mature core protein will be generated after further cleavage by signal peptide peptidase (empty arrowhead). The NS2/NS3 junction site is cleaved by the NS2-NS3 auto-protease, and the remaining nonstructural proteins are processed by the NS3/4A proteinase. (**c**) All of the HCV proteins are directly or indirectly associated with the endoplasmic reticulum. Currently used anti-HCV direct-acting antivirals (DAAs) target NS3, NS5A, and NS5B, respectively. NS3, NS4A, NS4B, NS5A, and NS5B proteins will form the replication complex.

**Figure 3 viruses-13-00520-f003:**
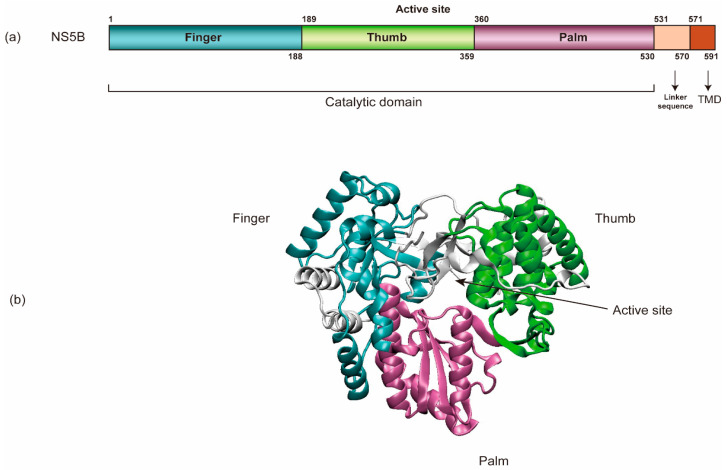
The schematic organization of NS5B’s structure. (**a**) NS5B encompasses an amino-terminal catalytic domain, a linker sequence, and a C-terminal transmembrane domain (TMD). (**b**) NS5B’s catalytic domain has a “right-hand” shape containing palm, thumb, and finger regions (PDB: 3FQK).

**Figure 4 viruses-13-00520-f004:**
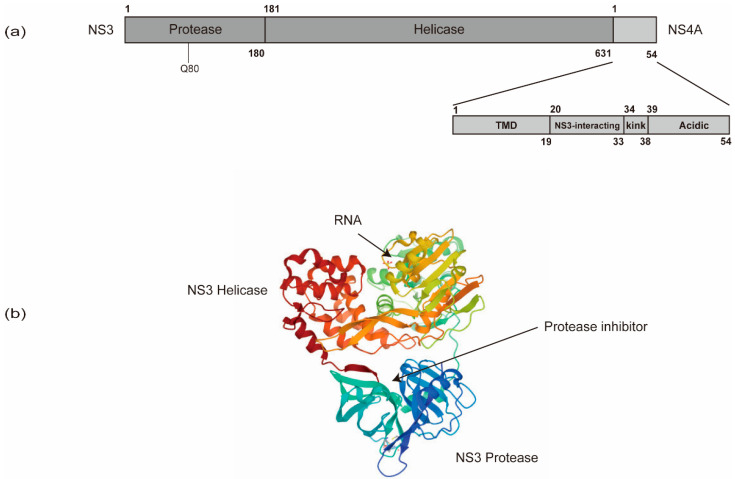
The schematic organization of NS3 and NS4A domains. (**a**) NS3 contains an amino-terminal protease domain and a carboxy-terminal DExD-box helicase domain. (**b**) The NS3 protease domain is responsible for HCV polyprotein processing and thus serves as a target for direct-acting antivirals (DAAs). The resistance-associated variants (RAV) of protease inhibitors are often in Q80. The NS3 helicase domain is responsible for HCV RNA replication via the unwinding of RNA secondary structures (PDB: 308B). NS4A contains an N-terminal transmembrane domain (TMD), a central NS3-interacting domain, and a C-terminal domain with a kink region and an acidic region.

**Figure 5 viruses-13-00520-f005:**
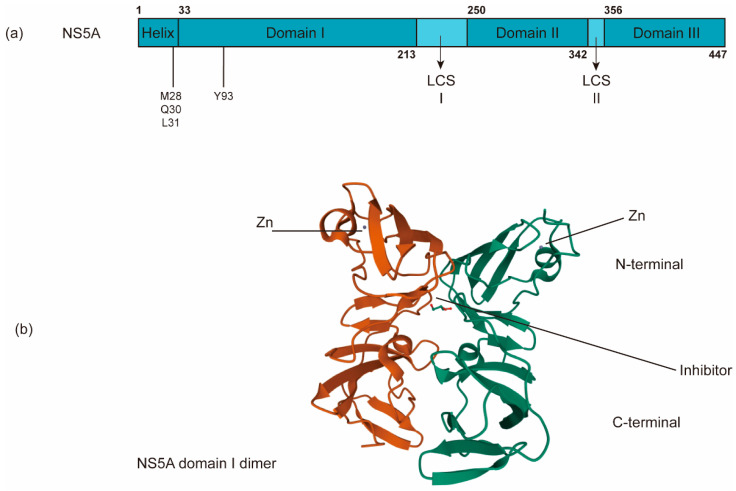
The schematic organization of NS5A domains. (**a**) NS5A contains an N-terminal amphipathic α-helix critical for its targeting of the membrane and three domains (domains I, II and III) separated by two low-complexity sequences (LCS). (**b**) NS5A inhibitors bind the dimerized domain I, thereby potentially blocking NS5A to exert its functions (PDB: 3FQQ). RAVs of NS5A inhibitors are often in M28, Q30, L31, and Y93.

**Table 1 viruses-13-00520-t001:** Hepatitis C virus (HCV) proteins play different roles in viral replication.

Viral Protein	Role in HCV Replication
Core	Package HCV genomic RNA to form nucleocapsids and also involve in lipid synthesis
E1, E2	Responsible for the entry of virions to cells
p7	Ion channel
NS2	Auto-protease to cleave the junction between NS2 and NS3
NS3	NS3 contains an amino-terminal protease domain responsible for the HCV polyprotein processing and a carboxy-terminal DExD-box helicase domain responsible for HCV RNA replication through unwinding RNA secondary structures
NS4A	Cofactor for NS3 protease
NS4B	To serve as a scaffold for the viral replication complex and to induce the rearrangements of membrane vesicles
NS5A	To interact with a large number of cellular proteins that are important for viral assembly and function of the replication complex
NS5B	HCV RNA-dependent–RNA-polymerase responsible for HCV RNA amplification

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
