# Peer review of "Hepatitis C Viral Replication Complex"

_viruses, 2021, doi:10.3390/v13030520_

Round 1

Reviewer 1 Report

The paper by Li et al. cites numerous other papers and clearly explains the matters it addresses. It’s an interesting review paper on an important topic. However, there are several points that should be addressed to further strengthen it.

Minor point:

Line 59: Li et al. says that DMVs constituted the predominant HCV-induced membrane structure that accumulated in the cytoplasm close to the lipid droplets in cultured cells. Lipid droplets are important organelles for HCV particle formation and are located near the replication complex. The focus of this review is “replication,” but I would like to ask you to give a short explanation of the role that lipid droplets play between replication and particle formation.

Line 85: Li et al. says that DMVs conferred several advantages for the formation of the viral ROs for the HCV RNA synthesis. The fifth advantage is as follows: Several reports show that viral RNA and proteins associated with the viral ROs are protected from cellular proteases and nucleases, indicating that RNA replication occurs in a membranous environment separated from the surrounding cytoplasm (line 72).

Line 301: In addition, sphingolipids (e.g., sphingomyelin) were shown to stimulate HCV replication activity. Sphingomyelin and cholesterol also play important roles in DMV formation. Please introduce the paper (Gewaid H et al., J Virol. 2020) and explain the role of sphingomyelin/cholesterol in DMV formation.

Author Response

Reviewer 1

Comments and Suggestions for Authors

The paper by Li et al. cites numerous other papers and clearly explains the matters it addresses. It’s an interesting review paper on an important topic. However, there are several points that should be addressed to further strengthen it.

Response: Thanks for your constructive suggestions. These suggestions make the manuscript much better.

Minor point:

Line 59: Li et al. says that DMVs constituted the predominant HCV-induced membrane structure that accumulated in the cytoplasm close to the lipid droplets in cultured cells. Lipid droplets are important organelles for HCV particle formation and are located near the replication complex. The focus of this review is “replication,” but I would like to ask you to give a short explanation of the role that lipid droplets play between replication and particle formation.

Response: Thanks for the suggestion. A short explanation of LDs is added as suggested in the first section of page 3 marked in red.

Line 85: Li et al. says that DMVs conferred several advantages for the formation of the viral ROs for the HCV RNA synthesis. The fifth advantage is as follows: Several reports show that viral RNA and proteins associated with the viral ROs are protected from cellular proteases and nucleases, indicating that RNA replication occurs in a membranous environment separated from the surrounding cytoplasm (line 72).

Response: Thanks for the suggestion. The sentence is added as suggested in page 3 marked in red.

Line 301: In addition, sphingolipids (e.g., sphingomyelin) were shown to stimulate HCV replication activity. Sphingomyelin and cholesterol also play important roles in DMV formation. Please introduce the paper (Gewaid H et al., J Virol. 2020) and explain the role of sphingomyelin/cholesterol in DMV formation.

Response: Thanks for the suggestion. The sentence is added as suggested in page 8 marked in red.

Reviewer 2 Report

Authors described the replication cycle of HCV virus and the RNA replication in the replication organelles. The viral and cellular factors involved in regulating HCV was discussed and the targets of the anti-viral therapy against HCV. The review provides an interesting summary of already published data, however the following items should be addressed:

  • Review should be revised thoroughly for language and grammar mistakes
  • Lines 31 and 37; authors state (for a review..). this is supposed to be a review of the HCV replication, so either authors summarize the data related to the topic or just indicate the reference.
  • The introduction should indicate the geographical distribution of the main HCV genotypes and countries with highest prevalence
  • Authors focused on the Viral replication organelles (RO) and double-membrane vesicles 56
  • (DMVs) induced by HCV infection however ignored the entry step of HCV replication. Authors should illustrate the receptors utilized by HCV for cellular binding which is the initial step of replication.
  • I suggest inserting a table that contains the different HCV proteins (structural and non-structural) and role of each protein in HCV replication. This will make it easier for the readers to follow the importance of each viral component in multiplication.
  • Line 188: it is not suitable to place the reference on the title
  • Line 341: authors described the role of DAA in treatment of HCV. Why authors did not consider discussing the role of interferon and ribavirin therapy.

Author Response

Reviewer 2

Comments and Suggestions for Authors

Authors described the replication cycle of HCV virus and the RNA replication in the replication organelles. The viral and cellular factors involved in regulating HCV was discussed and the targets of the anti-viral therapy against HCV. The review provides an interesting summary of already published data, however the following items should be addressed:

Response: Thanks for your constructive suggestions. These suggestions make the manuscript much better.

  • Review should be revised thoroughly for language and grammar mistakes

Response: The entire manuscript has been edited by MDPI editing service (certificate 27934).

  • Lines 31 and 37; authors state (for a review..). this is supposed to be a review of the HCV replication, so either authors summarize the data related to the topic or just indicate the reference.

Response: (for a review..). in the entire manuscript has been deleted as suggested.

  • The introduction should indicate the geographical distribution of the main HCV genotypes and countries with highest prevalence

Response: The geographical distribution of the main HCV genotypes has been added as suggested in the first section of page 1 marked in red.

  • Authors focused on the Viral replication organelles (RO) and double-membrane vesicles (DMVs) induced by HCV infection however ignored the entry step of HCV replication. Authors should illustrate the receptors utilized by HCV for cellular binding which is the initial step of replication.

Response: The receptors utilized by HCV for cellular binding has been added as suggested in the last section of page 1 marked in red.

  • I suggest inserting a table that contains the different HCV proteins (structural and non-structural) and role of each protein in HCV replication. This will make it easier for the readers to follow the importance of each viral component in multiplication.
  • Response: A table has been added as suggested in the page 2 marked in red.
  • Line 188: it is not suitable to place the reference on the title
  • Response: Reference on the title has been deleted as suggested.
  • Line 341: authors described the role of DAA in treatment of HCV. Why authors did not consider discussing the role of interferon and ribavirin therapy.

Response: The role of interferon and ribavirin therapy has been added as suggested in the page 9 marked in red

Round 2

Reviewer 2 Report

The authors modified the manuscript according to my suggestions. In addition, the language and grammar were corrected by MDPI. I do not have any further comments.